♿ | **Open Peer Review** | Bacteriology | Resource Report

# Construction of an arrayed CRISPRi library as a resource for essential gene function studies in *Streptococcus mutans*

Jackson St. Pierre,[1,2] Justin Roberts,[3,4] Mohammad A. Alam,[3] Robert C. Shields[1]

**ABSTRACT** Arrayed mutant libraries are important resources that have advanced our understanding of bacterial genetics. Until recently, essential genes which are necessary for bacterial survival, and constitute ~10% of bacterial genes, were not included in arrayed library resources. However, recent advances in clustered regularly interspaced short palindromic repeats interference (CRISPRi) have made it possible to study essential genes with increasing throughput. Here, we constructed an arrayed CRISPRi library in the dental caries pathogen *Streptococcus mutans*, known as SNAP (***S**treptococcus mutans* **a**rrayed CRIS**P**Ri). In this library, each strain contains a short guide RNA which selectively targets a single essential gene, and this is controlled through xylose induction. In total, the library can selectively repress >250 essential and growth-supporting genes. Initial characterization showed that SNAP strains grow as expected, and initial proof-of-concept experiments displayed the utility of the library. We anticipate that this library will be of benefit to the research community by allowing for high-throughput characterization of *S. mutans* essential genes.

**IMPORTANCE** The construction of arrayed mutant libraries has advanced the field of bacterial genetics by allowing researchers to more efficiently study the exact function and importance of encoded genes. In this study, we constructed an arrayed clustered regularly interspaced short palindromic repeats interference (CRISPRi) library, known as ***S**treptococcus mutans* **a**rrayed CRIS**P**Ri (SNAP), as a resource to study >250 essential and growth-supporting genes in *Streptococcus mutans*. SNAP will be made available to the research community, and we anticipate that its distribution will lead to high-quality, high-throughput, and reproducible studies of essential genes.

**KEYWORDS** *Streptococcus mutans*, CRISPRi, functional genomics, arrayed mutant library, essential genes

Address correspondence to Robert C. Shields, rshields@astate.edu.

The authors declare no conflict of interest.

See the funding table on p. 9.

Bacterial genetics research has been greatly accelerated by arrayed mutant library resources. Examples include the *Escherichia coli* Keio collection (1), the *Staphylococcus aureus* USA300 NTML transposon mutant library (2), the *Streptococcus sanguinis* SK36 deletion mutant library (3), and *Enterococcus faecalis* OG1RF SmarT (4, 5). These libraries are focused on nonessential genes that are inactivated by gene deletion or transposon insertion, and they have yielded important insights into gene functions across diverse environments (6–8). Recent advances have allowed bacterial genetics researchers to study an important fraction of genes that are not present in these libraries. These genes, known as "essential genes," are required for survival and, thus, are not amenable to deletion via transposon mutagenesis or gene replacement. CRISPR (clustered regularly interspaced short palindromic repeats) interference, a molecular tool that facilitates the targeted repression of genes, has been used to study essential genes in several microorganisms (9). These studies have used collections of CRISPRi strains, arrayed libraries, and genome-wide approaches (10–13). This has greatly accelerated

the study of essential genes leading to characterization of unknown essential genes and has highlighted interactions between critical essential pathways (14, 15). Recently, we developed a CRISPRi tool in the Gram-positive dental caries pathogen *Streptococcus mutans* (16). The system targets the repression of more than 250 essential and growth-supporting genes. Collecting these strains and arraying them into a library format would speed up the study of essential genes in *S. mutans* in a way that allows for efficient and robust replication of the library as a resource for other researchers.

Here, we describe the design, construction, and proof-of-concept testing of an arrayed CRISPRi library in *S. mutans* known as SNAP (**S**treptococcus mutans **a**rrayed CRIS**P**Ri). The mutants are derivatives of strain UA159, which was originally isolated from a child with dental caries and is widely used as a model *S. mutans* strain. CRISPRi strains were transformed using previously generated sgRNA (single guide RNA) plasmids (16) and then placed into a 96-well format for long-term ultracool storage. After constructing the SNAP library, we describe validation studies comparing growth of the SNAP library to previously generated growth data with the same CRISPRi strains. Confident the arrayed CRISPRi library was operating as expected, we then profiled the library in clinically relevant growth conditions as proof of concept for the utility of the library. New replicates of SNAP, which consists of three 96-well plates, are generated quickly using replicating pins and will be made available to the research community for genotype-phenotype testing and other mutant screening approaches.

## RESULTS

### Construction of an arrayed CRISPRi library in *Streptococcus mutans*

We previously constructed ~250 plasmids with each one containing a single sgRNA which when expressed in conjugation with dCas9 leads to repression of a single essential or growth-supporting gene. Our basis for targeting these genes with a CRISPRi system was conceived from prior transposon sequencing (Tn-seq) efforts exploring gene essentiality in *S. mutans* (17). Validation of this CRISPRi system included determining repression with green fluorescent protein (GFP) readouts, measuring the effect of CRISPRi on the *S. mutans* transcriptome, and investigating sgRNA target-binding rules (16). When designing each sgRNA, care was taken to avoid potential off-target effects that are known to occur with CRISPR-based technologies (18). In our prior work, each CRISPRi plasmid/strain was maintained in separate storage tubes, and when needed, these strains would be grown individually. Although this is a suitable method for targeted assays, it is not amenable to phenotypic screening of the entire library in a time-efficient manner. In addition, it does not allow for providing each *S. mutans* CRISPRi strain to researchers quickly. To overcome these limitations, we constructed an arrayed CRISPRi library in microtiter plates. Briefly, we transformed each individual sgRNA-containing CRISPRi plasmid into *S. mutans* Δcas9 P$_{xyl}$-dcas9. It is important to state that our *S. mutans* UA159 strain (the backbone for all CRISPRi strains) is lab adapted and has a truncated *perR* gene similar to that reported by Kajfasz et al. (19). PerR is a transcriptional regulator that responds to oxidative stress, and the mutation of *perR* in our lab-adapted strain of *S. mutans* likely confers an advantage in aerobic conditions. Each sgRNA targets the 5′ region of essential or growth-supporting genes. Repression of genes with CRISPRi will also silence downstream genes (polar effects), and phenotypic results should be evaluated with this in mind. The sgRNA element integrates within a non-essential region of the *S. mutans* genome (between *phnA* and *mtlA*), and the d*cas9* is carried on a plasmid (derivative of pDL278). The d*cas9* is controlled by the P$_{xyl}$ promoter, which becomes active when xylose is added to the growth medium. Our previous work has shown that xylose allows for titratable repression of genes up to ~95% knockdown (tested with *gfp* targeting) (16). After transformation, individual CRISPRi strains were arrayed into single wells of microtiter plates (see Materials and Methods). In total, the SNAP library consists of three 96-well microtiter plates, with 270 strains (including controls) arrayed across the plates. Each plate contains control strains in wells A1, B1, and C1 that should not have growth defects under most conditions. These control strains are

*S. mutans* Δcas9 P$_{xyl}$-dcas9 without an sgRNA, a CRISPRi strain targeting *gtfB* (important for sucrose-mediated biofilm formation), and a CRISPRi strain targeting *scrB* (required for sucrose utilization). The layout of each microtiter plate is shown in Fig. S1 in the supplementary material.

Following the generation of the SNAP library platform, we began by testing the growth of arrayed strains under control conditions. First, the library was grown in rich media (brain heart infusion broth). Plates were incubated in a microplate reader (Synergy 2) at 37°C for 16 h, with OD$_{600}$ measurements recorded every 30 min. Afterward, the area under the curve (AUC) was calculated for each well and compared to the average AUC of control strains. For our purposes, an AUC$_{sgRNA}$/AUC$_{control}$ of ≤−1 represents a significant growth defect compared to control strains (with a *P*-value <0.05; as determined with a Welch's *t*-test). As expected, no CRISPRi strains exhibited significant growth defects under control conditions (Fig. 1A). Next, the arrayed library was cultured in defined media containing 0.1% xylose. Xylose is the inducing molecule that causes the expression of dCas9 in our CRISPRi strains. Under these conditions, a significant number (141) of strains exhibited growth defects, consistent with previous analysis of these strains (Fig. 1B; see Table S1 for AUC$_{sgRNA}$/AUC$_{control}$ raw data) (16). In the original study, CRISPRi strains were cultured using a Bioscreen C automated growth curve system using the defined media known as FMC (20, 21). In this study, strains were cultured using a microplate reader and a chemically defined media with a different composition to FMC (22). Our goal when choosing the new conditions was to improve standardization, as more labs have access to microplate reader systems, and the "CDM" recipe is more widely used than FMC. New copies of SNAP library plates are quickly made using replicating pins. To reduce contamination, cold-induced killing, and genetic drift that may be caused by repeat freeze-thaw cycles, SNAP library plates are used a maximum of five times before being discarded.

## Profiling the impact of essential gene repression on biofilm formation

For a preliminary proof-of-concept study, we wanted to test if biofilm accumulation is disrupted similar to planktonic growth when essential genes are repressed by CRISPRi. Biofilm accumulation, particularly in the presence of sucrose, is a major virulence determinant of *S. mutans* (23). For this study, the arrayed CRISPRi library was screened in CDM containing 20 mM glucose and 10 mM sucrose with 0.25%, 0.5%, and 1% xylose.

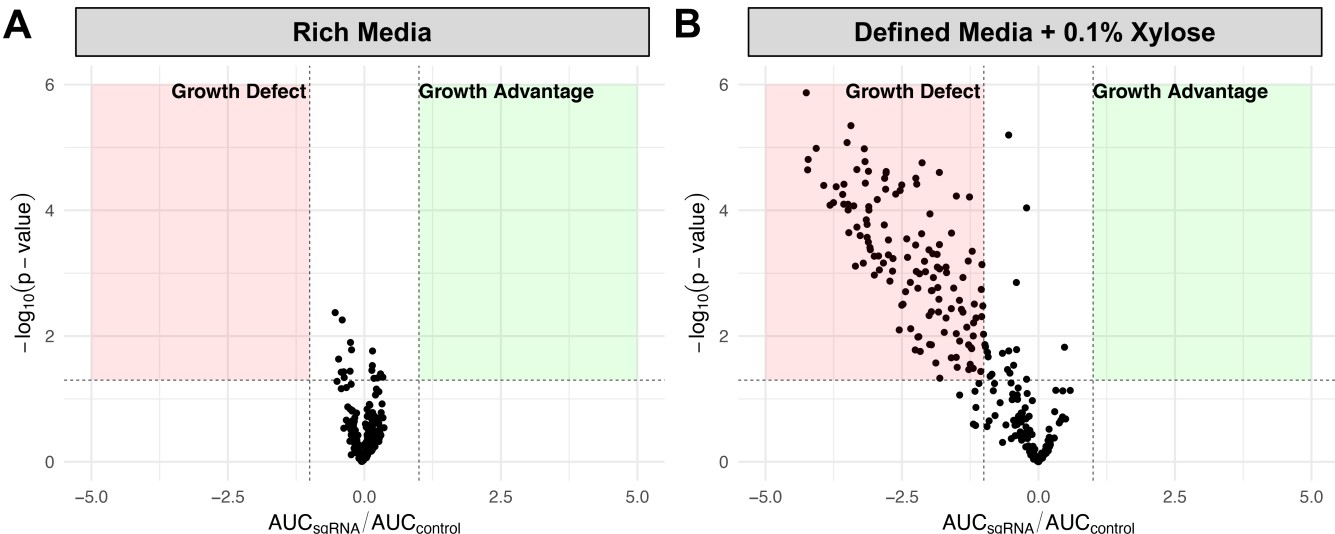

**FIG 1** SNAP library reproduces expected growth phenotypes. The SNAP library was grown in rich media (BHI) (A) and CDM supplemented with 0.1% xylose (B). For each strain, the area under the curve (AUC) was calculated and compared to the average AUC of control strains. If AUC$_{sgRNA}$/AUC$_{control}$ is ≤−1, this represents a significant growth defect compared to control strains (with a *P*-value <0.05). Each condition was repeated in triplicate, and each dot represents the mean value of the triplicates.

Biofilm accumulation of the control strains was as expected (Fig. 2A). The addition of 0.5% and 1% xylose led to a significant ($P < 0.05$) decrease in biofilm accumulation when *gtfB* was targeted. This is expected because extracellular polysaccharides produced by cell-associated glycotransferases (GtfB and GtfC) are a major determinant of sucrose-mediated biofilm formation in *S. mutans* (24). The impact of *gtfB* repression on biofilm accumulation was minor when the xylose concentration was reduced to 0.25%. Only minor reductions in biofilm accumulation were observed when targeting *scrB* and in the CRISPRi strain that lacks a sgRNA (Fig. 2A). Across the entire library, basal repression (no xylose addition) did not have a major impact on biofilm accumulation (Fig. 2B). However, strong repression with 0.5% xylose had a heterogenous impact on biofilm accumulation (Fig. 2B). Most SNAP library strains (57%; 147/258 strains) had a reduction in biofilm biomass when the essential or growth-supporting gene was repressed (defined as a>15% reduction in biofilm accumulation with xylose added). However, almost a quarter of the strains (21%; 55/258) had increased biofilm biomass upon xylose-mediated repression of gene expression (defined as a > 15% increase in biofilm accumulation with xylose added). These data highlight nuanced effects of CRISPRi repression of essential genes on biofilm accumulation. The data also confirm that the SNAP library can be used for high-throughput screening of the phenotypic impacts of essential gene repression in different environments.

## Drug target discovery with a CRISPRi screen

In *Bacillus subtilis*, basal repression of essential genes by CRISPRi has been used to discover the mechanism of action of new bioactive compounds (10). This works because dCas9 is produced without induction because of $P_{xyl}$ promoter leakiness, leading to slight repression (~3-fold) of essential genes. Our CRISPRi system also uses $P_{xyl}$ for dCas9 induction, and it has been reported to have high basal activity (25). To explore if leakiness of the $P_{xyl}$ promoter was measurable, we compared green fluorescent protein (GFP) production in *S. mutans* Δcas9 $P_{xyl}$-dcas9 versus a strain also containing sgRNA[gfp]. Without xylose present there was a reduction in GFP production of ~20%, as

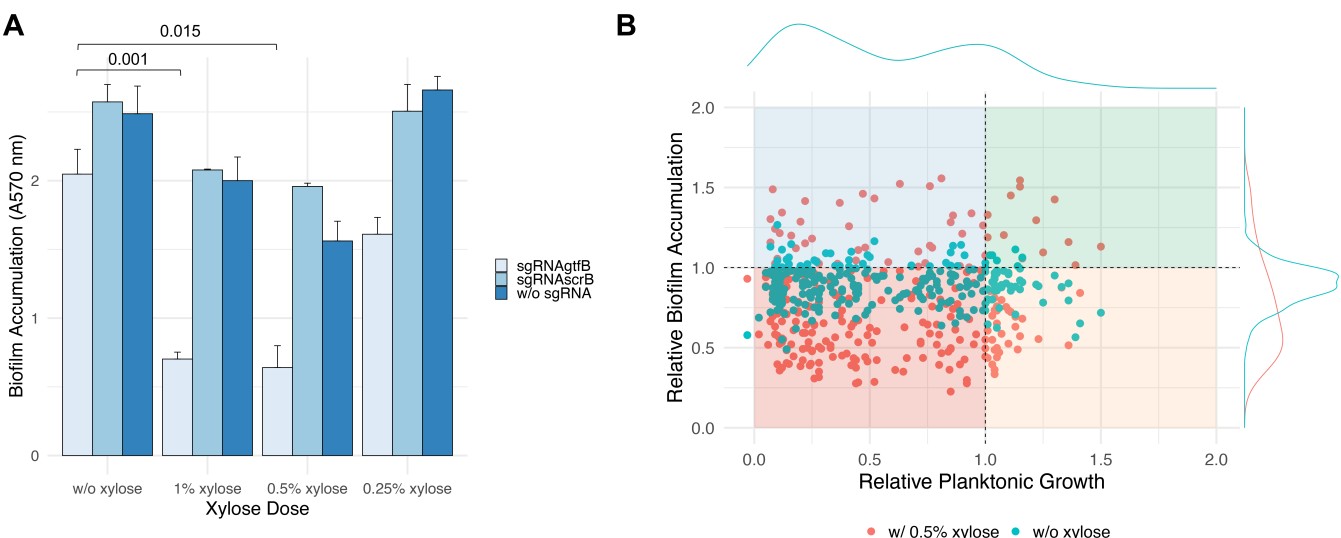

**FIG 2** Heterogenous impact of essential gene repression on sucrose-mediated biofilm accumulation. (A) SNAP library control strains were cultured in CDM containing 20 mM glucose and 10 mM sucrose, and biofilm accumulation was quantified using the crystal violet assay. Statistical significance was calculated using the Welch's *t*-test. Only statistically significant comparisons are shown; all other comparisons were found to be not significant. Error bars represent one standard error from three different experiments. (B) Scatter plot showing relative planktonic growth (with xylose) compared to relative biofilm accumulation (without xylose and with 0.5% xylose). The distribution of the numeric values is shown with density plots. Each quadrant represents a different phenotype compared to control strains: red is low planktonic growth and low biofilm accumulation, blue is low planktonic growth and high biofilm accumulation, orange is high planktonic growth and low biofilm accumulation, and green is high planktonic growth and high biofilm accumulation. Each condition was repeated in triplicate, and each dot represents the mean value of the triplicates.

measured by fluorescence, in the strain with sgRNA$^{gfp}$ present (Fig. S2). With the addition of 0.1% xylose, GFP production was repressed by ~95%. Next, we wanted to test if slight repression of genes, because of P$_{xyl}$ basal activity, could sensitize *S. mutans* to antimicrobials. For our proof-of-concept studies, we designed CRISPRi strains that target genes required for survival in the presence of sulfamethoxazole (SMX) and penicillin G (PenG). SMX targets folic acid synthesis by interfering with dihydropteroate synthase, and PenG binds to transpeptidases (penicillin-binding proteins) that are essential for cell wall synthesis. As shown in Fig. S3A, CRISPRi targeting of *folP* (dihydropteroate synthase) increased sensitivity of *S. mutans* to SMX compared to a control strain targeting the carbohydrate metabolism gene *lacG*. Targeting *pbp2b* with CRISPRi increased *S. mutans* sensitivity to PenG (Fig. S3B). Our data suggest that the arrayed CRISPRi system could be used to identify essential genes that are sensitive to antimicrobials. Note, there is a contrast in that basal repression does not cause growth defects in rich media (Fig. 1A) but the addition of sub-MIC antibiotics does. In *E. coli* and *B. subtilis,* it has been suggested that essential gene products are produced in sufficiently large quantities that despite basal repression, in rich media, cells are buffered against significant deleterious effects (10, 11). However, the addition of antimicrobials that target specific essential genes/pathways causes this buffer to be disrupted, and the result is a loss of growth.

After our preliminary studies, we next focused on testing the arrayed CRISPRi library in the presence of sub-MIC concentrations of two oral healthcare products, sodium fluoride (NaF), and cetylpyridinium chloride (CPC). NaF is added to toothpaste and municipal drinking water because it supports the remineralization of teeth after dissolution caused by acidogenic bacteria (26). It also has mild antimicrobial activities when added at sufficient concentrations (27). *S. mutans* possess transporter mechanisms (*perAB*) to protect against this activity (28–30). The activities of pyruvate kinase, enolase, and the F-ATPase are also reportedly sensitive to NaF (28, 29, 31). Before screening, *S. mutans* growth in the presence of NaF was measured across a range of concentrations (31.25–500 µg/mL) and 125 µg/mL was chosen as a sub-MIC (Fig. S4A). When the arrayed library was grown in the presence of NaF, none of the 270 CRISPRi strains tested exhibited any significant growth defects (Fig. 3A; see Table S1 for AUC$_{sgRNA}$/AUC$_{control}$ raw data). This suggests that at a concentration of 125 µg/mL, there are no essential genes in our library that are sensitive to NaF. Pyruvate kinase (Plate 1, *pykF*), enolase (Plate 1, *eno*), and components of the F-ATPase (Plate 1, *atpF, atpC, atpA, atpD*; Plate 3,

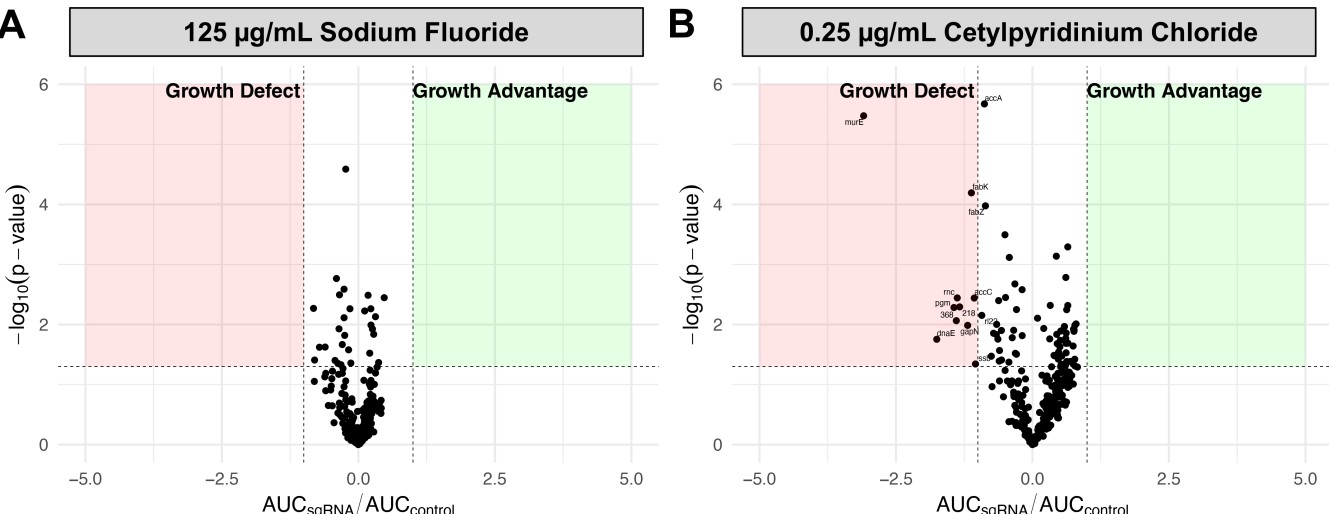

**FIG 3** SNAP library indicates that the cell envelope is targeted by the antimicrobial agent CPC. The SNAP library was grown in rich media (BHI) with sub-MIC concentrations of sodium fluoride (NaF) (A) and the quaternary ammonium compound CPC (B). For each strain, the area under the curve (AUC) was calculated and compared to the average AUC of control strains. If AUC$_{sgRNA}$/AUC$_{control}$ is ≤−1, this represents a significant growth defect compared to control strains (with a *P*-value <0.05). Each condition was repeated in triplicate, and each dot represents the mean value of the triplicates.

*atpB*) are in our CRISPRi library. Although NaF has been reported to alter the activity of these enzymes, our findings suggest that basal repression of these genes does not impact NaF tolerance at 125 µg/mL. Testing with a higher concentration of NaF (~200 µg/mL) might increase the sensitivity of CRISPRi strains to NaF. It is also possible that the repression of more than one essential gene at the same time (in a multiplex format) might cause more significant defects.

Next, the SNAP library was screened against sub-MIC amounts of CPC. Cetylpyridinium chloride is a broad-spectrum cationic quaternary ammonium compound (QAC) that has rapid bactericidal effects (32). Positively charged CPC interacts with the bacterial cell membrane causing cell lysis and leakage of cytoplasmic proteins (32). During preliminary growth assays, a sub-MIC amount of 0.25 µg/mL was selected (Fig. S4B). Screening against CPC revealed several CRISPRi strains that exhibited increased sensitivity to the compound (Fig. 3B; see Table S1 for $AUC_{sgRNA}/AUC_{control}$ raw data). In total, 10 CRISPRi strains targeting the following essential genes met our criteria for sensitivity ($AUC_{sgRNA}/AUC_{control}$ is $\leq -1$; *P*-value $<0.05$): *murE*, *fabK*, *accC*, *pgm*, *dnaE*, *rnjA*, *rnc*, SMu.218 [$immR_{Smu}$, putative TnSmu1 regulator (33, 34)], *gapN*, and *ssb*. Of particular interest were CPC sensitive CRISPRi strains involved in cell wall synthesis (*murE* and *pgm*) and lipid metabolism (*accC* and *fabK*), including an additional two strains (*accA* and *fabZ*) that were close to our criteria for increased sensitivity. These results are consistent with CPC disrupting the *S. mutans* cell envelope, which would be the expected mechanism of action for the compound. We confirmed the six strains sensitivity to CPC by re-cloning the CRISPRi strains and repeating growth assays (Fig. 4). The results were similar to the

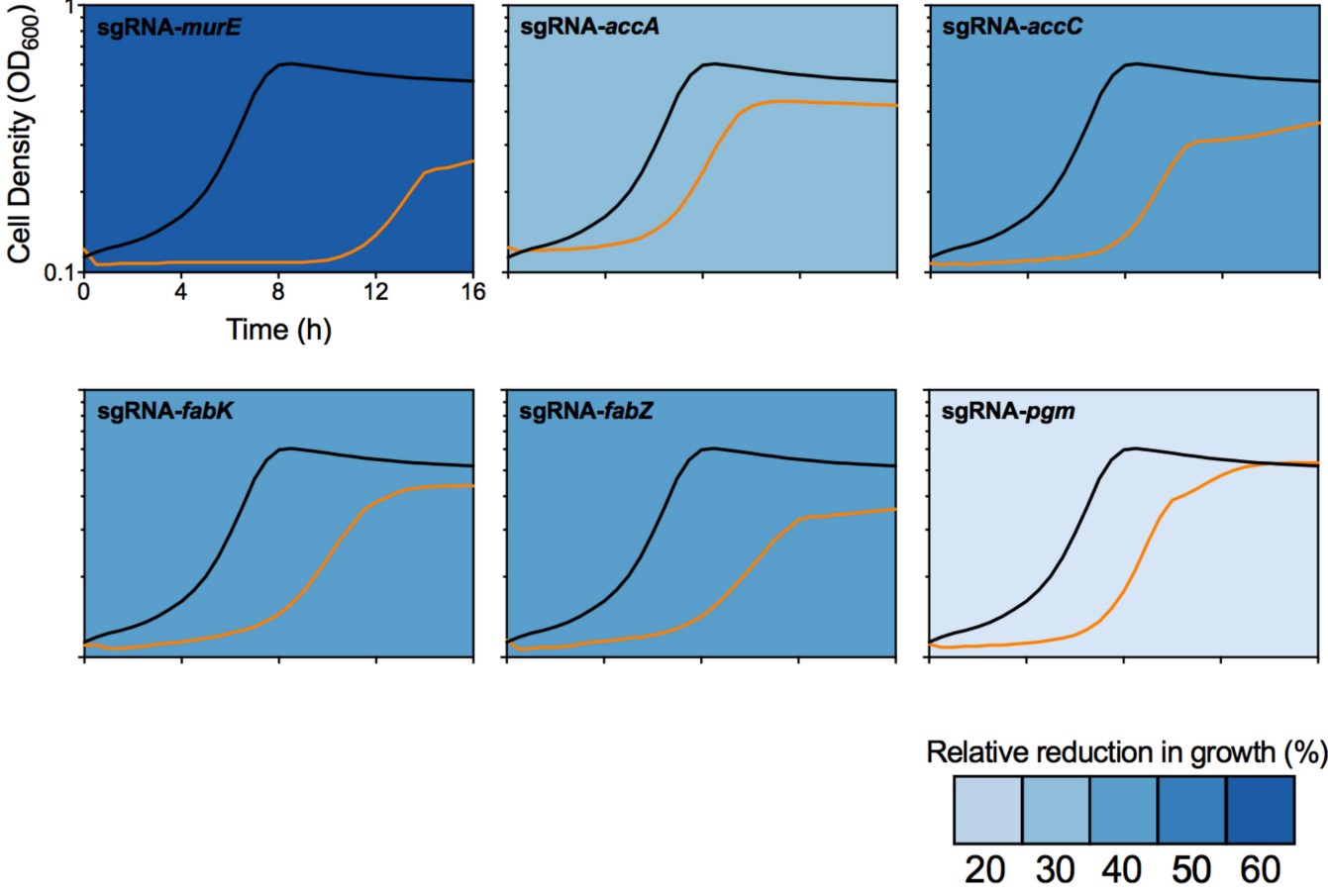

**FIG 4** Basal repression of cell envelope-associated genes increases sensitivity to CPC. After SNAP library screening, CRISPRi strains were re-transformed and grown in the presence of 0.25 µg/mL CPC (sub-MIC). The growth curve of CRISPRi strains is shown in orange and the control in black. Growth curves are representative of three biological replicates.

arrayed screens, with sgRNA$^{murE}$ most sensitive to CPC treatment (Fig. 4). We anticipate that increased sensitivity to CPC in the above strains is likely caused by an increased propensity for these strains to lyse in the presence of CPC. CRISPRi strains not directly related to cell envelope biogenesis also exhibited increased sensitivity to CPC (e.g., *dnaE* and *rnjA*). Morphological examination of these strains in previous work revealed robust morphological defects (16). We reasonably expect that these CRISPRi strains might also have cell envelope defects that increase CPC sensitivity.

## DISCUSSION

Arrayed collections of bacterial strains are valuable resources that increase the through-put and reproducibility of gene function screens. These libraries have targeted non-essential genes (e.g., the *E. coli* KEIO collection). However, recent innovations with CRISPRi have made it possible to target all essential genes in arrayed libraries. Examples of arrayed CRISPRi libraries targeting essential genes are those generated for *Mycobacterium smegmatis* (14), *E. coli* (11, 35) and *B. subtilis* (10). Using these libraries, researchers have been able to screen morphological phenotypes, assess bulk growth phenotypes, characterize novel essential genes, and determine the mechanism of action (MoA) of antimicrobials. With the success of these recent approaches, and building on from a recent study where we designed and constructed the CRISPRi tool in *S. mutans* (16), we constructed SNAP. Initial library validation showed expected growth profiles in both rich media and when CRISPRi gene repression was induced with xylose. After the initial validation, we used the library for two proof-of-concept high-throughput functional screens: (i) assessing the contribution of essential genes to biofilm formation and (ii) determining the MoA for CPC. These two screens demonstrate the potential utility of the SNAP library as a method for rapidly assessing the role of essential *S. mutans* genes in specific environments. The SNAP library has also been utilized to determine the mode of action of novel pyrazole derivatives against *S. mutans* (36). The library's format, which is arranged into three 96-well plates, makes it simple to replicate, and it will be made available to researchers upon request.

## MATERIALS AND METHODS

### Bacterial strains and culture conditions

For routine growth, *S. mutans* was cultured at 37°C with a 5% $CO_2$ atmosphere in brain heart infusion broth (BHI; Difco). When necessary, the growth medium was supplemented with antibiotics, 1 mg/mL spectinomycin and 1 mg/mL kanamycin. *E. coli* cells were grown with aeration at 37°C in Luria-Bertani broth (LB; Lennox formula; Fisher), and when necessary, LB medium was supplemented with 50 µg/mL kanamycin.

### Arrayed CRISPRi library construction

CRISPRi strains were constructed as previously described (16). Briefly, sgRNA plasmids were transformed into *S. mutans* Δcas9 P*xyl*-dcas9 (16) using competence stimulating peptide (CSP). Transformation mixtures were plated onto double-selection plates (containing kanamycin and spectinomycin). Single colonies from each transformation were then subcultured into deep-well microplates containing 1 mL BHI plus antibiotics. Deep-well plates were sealed with Axygen microplate sealing films (Corning). After overnight growth, 100 µL of each strain was transferred to a single well of a 96-well plate that contained 100 µL 50% glycerol (for a final concentration of 25% glycerol). In total, 270 CRISPRi strains (including controls) were arrayed across three 96-well microtiter plates before storage at −80°C. A plate layout diagram is shown in Fig. S1;A1 through C1 contain control strains that are non-essential in most conditions.

## Replication of the arrayed CRISPRi library

Frozen 96-well SNAP library plates were thawed until liquid and then an ethanol-sterilized replicator pin tool (Boekel Scientific) was used to transfer microliter amounts of strains into pre-filled deep-well plates. The deep-well plate was filled with BHI plus antibiotics, and after strain transfer, they were sealed with Axygen sealing films. After overnight culture, 100 µL of each replicated strain was transferred from the deep-well plate to a single well of a 96-well plate that was pre-filled with 100 µL 50% glycerol (for a final concentration of 25% glycerol). When making replicates of the SNAP library, we thaw "master" plates that are minimally used and do not undergo freeze-thaw cycles, to reduce the effect that this might have on SNAP library strains.

## Biofilm formation assay

To assay for biofilm formation, the SNAP library was cultured in chemically defined media (CDM) containing 20 mM glucose and 10 mM sucrose with a range of xylose concentrations. Before SNAP library transfer, polystyrene 96-well microplates (Falcon non-treated, flat bottom) were filled with 200 µL CDM (with or without xylose). Next, a sterilized replicator pin was used to transfer microliter amounts of each CRISPRi strain to the pre-filled microplate. Plates were incubated for 18 h at 37°C with a 5% $CO_2$ atmosphere. After incubation, the biofilm cultures were washed with phosphate-buffered saline (PBS) [137 mM NaCl, 2.7 mM KCl, 8 mM $Na_2HPO_4$, and 2 mM $KH_2PO_4$ (pH 7.4)] to remove loosely attached cells. Then, the biofilm cells were stained with 0.1% crystal violet (CV), 100 µL per well, for 20 min at room temperature and then washed three times with PBS. CV was dried at room temperature for approximately 30 min, and then 200 µL of 7% acetic acid was transferred into each well of the 96-well plates to solubilize the crystal violet stain. Absorbance of the CV solution was measured at 570 nm using a BioTek Synergy 2 Plate-Reader and BioTek Gen 5 Software Edition 2.09. Statistical significance comparing control biofilm formation versus CRISPRi strains was calculated with a Welch's $t$-test.

## CRISPRi strain construction

sgRNA plasmids for sgRNA$^{pbp2b}$ and sgRNA$^{folP}$ were cloned as previously described (16). Briefly, sgRNA sequences (20-nt) were selected using CRISPy-web (pbp2bF: cgagggctgggtttagagctagaaatagc; pbp2bR: cggacaaattacatttattgtacaacacg; folPF: gctgccaacagttttagagctagaaatagc; folPR: aggagcggctacatttattgtacaacacg) (37). The 20-nt sgRNA sequences were then cloned into the pPM::sgRNA plasmid using the Q5 Site-Directed Mutagenesis kit (New England Biolabs). Putative plasmids were Sanger sequenced using the pRCS1seqR primer (aaaacagccaagctggagac).

## Antimicrobial efficacy testing

The SNAP library was thawed, and a sterilized replicator pin was used to transfer CRISPRi strains to 96-well polystyrene microplates. The plates were pre-filled with BHI and sub-MIC amounts of either sodium fluoride (125 µg/mL) or cetylpyridinium chloride (0.25 µg/mL). The sub-MIC concentrations were discovered using serial dilution and a Bioscreen C Pro automated growth platform (Fig. S4). After SNAP library inoculation, microplates were transferred to a BioTek Synergy 2 microplate reader and incubated at 37°C for 16 h, with 600 nm absorbance readings taken every 30 min. Using the growth data, the area under the curve (AUC) was calculated for each CRISPRi strain and compared to the average AUC of control strains (w/o sgRNA, sgRNA$^{scrB}$, sgRNA$^{gtfB}$). $AUC_{sgRNA}/AUC_{control}$ of ≤−1 represents a growth defect compared to control strains. Statistical significance of growth curve AUCs was calculated with a Welch's $t$-test.

## ACKNOWLEDGMENTS

This work was supported by the National Institute of Dental and Craniofacial Research (NIDCR) grant DE029882 (R.C.S.), a Colgate-Palmolive CARE award (R.C.S.), and the Arkansas INBRE program (NIGMS P20 GM103429) (R.C.S. and M.A.A.).

## AUTHOR AFFILIATIONS

[1]Department of Biological Sciences, Arkansas State University, Jonesboro, Arkansas, USA
[2]New York Institute of Technology College of Osteopathic Medicine, Jonesboro, Arkansas, USA
[3]Department of Chemistry & Physics, Arkansas State University, Jonesboro, Arkansas, USA
[4]University of Arkansas for Medical Sciences, Little Rock, Arkansas, USA

## AUTHOR ORCIDs

Mohammad A. Alam http://orcid.org/0000-0003-0258-3732
Robert C. Shields http://orcid.org/0000-0002-8823-9504

## FUNDING

| Funder | Grant(s) | Author(s) |
| --- | --- | --- |
| HHS \| NIH \| National Institute of General Medical Sciences (NIGMS) | GM103429 | Mohammad A. Alam |
| | | Robert C. Shields |
| HHS \| NIH \| National Institute of Dental and Craniofacial Research (NIDCR) | DE029882 | Robert C. Shields |

## AUTHOR CONTRIBUTIONS

Jackson St. Pierre, Data curation, Formal analysis, Investigation, Writing – original draft, Writing – review and editing | Justin Roberts, Data curation, Investigation, Methodology, Writing – review and editing | Mohammad A. Alam, Funding acquisition, Resources, Writing – review and editing | Robert C. Shields, Conceptualization, Investigation, Methodology, Project administration, Writing – original draft, Writing – review and editing

## ADDITIONAL FILES

The following material is available online.

### Supplemental Material

**Supplemental material (Spectrum03149-23-s0001.docx).** Fig. S1 to S4 and Table S1.

### Open Peer Review

**PEER REVIEW HISTORY (review-history.pdf).** An accounting of the reviewer comments and feedback.

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
