## [Reviewer comments · Microbiology Spectrum]

Microbiology Spectrum

Construction of an arrayed CRISPRi library as a resource for essential gene function studies in *Streptococcus mutans*

Jackson St. Pierre, Justin Roberts, Mohammad Alam, and Robert Shields

Corresponding Author(s): Robert Shields, Arkansas State University

Review Timeline:

Submission Date:	August 22, 2023
Editorial Decision:	September 18, 2023
Revision Received:	November 1, 2023
Accepted:	November 8, 2023

Editor: Jose Lemos

Reviewer(s): Disclosure of reviewer identity is with reference to reviewer comments included in decision letter(s). The following individuals involved in review of your submission have agreed to reveal their identity: Ping Xu (Reviewer #1); Jonathon L Baker (Reviewer #2)

Transaction Report:

DOI: <https://doi.org/10.1128/spectrum.03149-23>

September 18, 2023

Dr. Robert C. Shields
Arkansas State University
Department of Biological Sciences
Jonesboro, Arkansas 72467

Re: Spectrum03149-23 (Construction of an arrayed CRISPRi library as a resource for essential gene function studies in *Streptococcus mutans*)

Dear Dr.Shields,

Thank you for submitting your manuscript to Microbiology Spectrum. Your manuscript was revised by 2 experts in the field that were generally positive about this work. However, both reviewers made several suggestions that you should consider in order to improve clarity and overall quality of the manuscript. When submitting the revised version of your paper, please provide (1) point-by-point responses to the issues raised by the reviewers as file type "Response to Reviewers," not in your cover letter, and (2) a PDF file that indicates the changes from the original submission (by highlighting or underlining the changes) as file type "Marked Up Manuscript - For Review Only". Please use this link to submit your revised manuscript - we strongly recommend that you submit your paper within the next 60 days or reach out to me. Detailed instructions on submitting your revised paper are below.

Link Not Available

Sincerely,

Jose Lemos

Journals Department
Reviewer comments:

Reviewer #1 (Comments for the Author):

In this study, the authors constructed an arrayed CRISPRi library to study the essential genes-related function in growth, biofilm formation, and antibiotic resistance in *S. mutans*. The innovative method is promising in the discovery of potential drug targets and study essential gene functions that usually cannot be obtained. This work could be very useful. However, there are several weaknesses that should be addressed before allowing for publication.

Major comments:

1. Essential genes are genes that an organism requires to survive under specific conditions. In this study, the essential gene functions can be induced-terminated. The identified genes are growth-associated or growth-supporting but not essential since the genes are not completely knocked out, and the growth is not entirely forbidden. As the author also mentioned, the leaking function of CRISPRi may work to make the organism survive. Gene knock-out should be performed with several positive controls to confirm the results.
2. How can you evaluate the target-off effect of CRISPRi? Do the results are validated through gene expression or protein expression levels?
3. Line 39-40, one more reference could be added here (<https://www.ncbi.nlm.nih.gov/pmc/articles/PMC3216606/>).
4. Line 84, the function of perR gene in Kajfasz's study should be described in the introduction.
5. Line 106, the method to calculate the AUCsgRNA/AUCcontrol values and how to get the p-values should be stated. Multiple testing corrections of the p-values should be performed.
6. Line 110, please add an SI table to show the AUCsgRNA/AUCcontrol values of the 141 strains.
7. Line 133, are the comparisons shown in Fig 2A between w/o xylose and 0.5% (or 1%) xylose of the sgRNAgtfB strain? What are the p-values of the comparison between w/o xylose and 0.5% (or 1%) xylose in the sgRNAscrB and w/o sgRNA strains?
8. Line 138, please explain the rationale for the threshold setting as 15%. An outlier analysis could be performed in the w/o xylose data to figure out a better threshold. The same comment is for the '0.5 increase' at line 142.
9. What does each dot mean in Fig. 1 and 2B? Is that the mean or median value of a strain? How many replicates are there for each mutant? Please state the information in the legends.

Reviewer #2 (Comments for the Author):

In this study, the authors describe the construction of an arrayed CRISPRi library of *Streptococcus mutans* UA159 and do some fundamental characterization and proof-of-concept experiments. This library will be a useful tool for the field. Overall, the manuscript is well-written, logical, and the conclusions are supported by the results. The content of the article also fits nicely with the scope of mSpectrum. Improvements needed prior to publication are mainly minor in nature, but will significantly improve clarity and readability for the audience. These improvements are listed point by point below:

Line 65: there should be a comma after "expected"

Figure S1: The black text is nearly impossible to read in some of the dark backgrounds. Please make the background color lighter or more transparent and/or change the text to white in these cases to improve readability.

Figure 1: I would suggest adding a header/title above each volcano plot so the reader can easily and quickly figure out what they are looking at. The size of the text relative to the figure should also be increased for readability.

Figure 2: Again, the size of the text is way too small relative to the figure to be readable...I printed out the document and I can't read it at all...I have to zoom way in on the computer. Also, the text in all figures is pixelated and low quality when you do zoom in.

Figure 2: The only comparisons that are statistically significant the two shown with ***?? None of the others are (w/o sgRNA w/o xylose vs 0.5 xylose, for example)?? Please clarify this and make sure it is clear to the reader which comparisons in the figure are and are not statistically significant.

Lines 141-143: This point is a bit confusing. Please explain what you mean here a little better, this is with or without xylose and what do you mean "by expected"?

Figure S3: Again, please add titles to the graphs. In the legend, I believe the panel letter should come before the text describing the panel, not after (titles will help clear that up, in any case). The graph legends are also confusing...the concentrations are for the antimicrobial added? Or inducer? Please make all this more clear.

Figure S4A: The order of concentrations in the legend makes no sense, please rearrange so they are in order of increasing. Also please add titles. Also, these are in mM, while the main text discusses things in terms of $\mu\text{g/ml}$, please make them all consistent. Also, if by 125 mM you mean 125 $\mu\text{g/ml}$, this does appear to inhibit growth, and therefore is not a sub-MIC concentration, as described in the main text?

Figure S4B: So the concentration used in the main experiment was 0.25 $\mu\text{g/ml}$, but that is not one of the growth curves shown? Since there did appear to be a minor lag of growth with 0.15625 and certainly a lag with 0.3125, please explain how you justify calling 0.25 a sub-MIC concentration.

Line 186: Why not test with higher NaF concentration where you might expect to see more impact and determine which genes are affected/impacted?

Figure 3: Again, please increase the size of the text relative to the figure. Is 1742 fabK? fabK is called out in the main text on line 195, but doesn't appear to be labeled as such in this graph.

Line 240: add reference for Smu Δcas9 P_{xyl}-dcas9 strain.

General comment: Please speculate on why you do not see reduced growth with the basal repression. You spend Figure 1A showing that there isn't reduced growth without xyl, but then suggest transcription of many genes is likely repressed enough to increase sensitivity to antimicrobials. I think this is worth a sentence or two to reconcile.

Staff Comments:

Preparing Revision Guidelines

Please return the manuscript within 60 days; if you cannot complete the modification within this time period, please contact me. If you do not wish to modify the manuscript and prefer to submit it to another journal, please notify me of your decision immediately so that the manuscript may be formally withdrawn from consideration by Microbiology Spectrum.

In this study, the authors constructed an arrayed CRISPRi library to study the essential genes related function in growth, biofilm formation, and antibiotic resistance in *S. mutans*. The innovative method is promising in the discovery of potential drug targets and study essential gene functions that usually cannot be obtained. This work could be very useful. However, there are several weaknesses that should be addressed before allowing for publication.

Major comments:

1. Essential genes are genes that an organism requires to survive under specific conditions. In this study, the essential gene functions can be induced-terminated. The identified genes are growth-associated or growth-supporting but not essential since the genes are not completely knocked out, and the growth is not entirely forbidden. As the author also mentioned, the leaking function of CRISPRi may work to make the organism survive. Gene knock-out should be performed with several positive controls to confirm the results.
2. How can you evaluate the target-off effect of CRISPRi? Do the results are validated through gene expression or protein expression levels?
3. Line 39-40, one more reference could be added here (<https://www.ncbi.nlm.nih.gov/pmc/articles/PMC3216606/>).
4. Line 84, the function of *perR* gene in Kajfasz's study should be described in the introduction.
5. Line 106, the method to calculate the $AUC_{sgRNA}/AUC_{control}$ values and how to get the p-values should be stated. Multiple testing corrections of the p-values should be performed.
6. Line 110, please add an SI table to show the $AUC_{sgRNA}/AUC_{control}$ values of the 141 strains.
7. Line 133, are the comparisons shown in Fig 2A between w/o xylose and 0.5% (or 1%) xylose of the sgRNA_{gtfB} strain? What are the p-values of the comparison between w/o xylose and 0.5% (or 1%) xylose in the sgRNA_{scrB} and w/o sgRNA strains?
8. Line 138, please explain the rationale for the threshold setting as 15%. An outlier analysis could be performed in the w/o xylose data to figure out a better threshold. The same comment is for the '0.5 increase' at line 142.
9. What does each dot mean in Fig. 1 and 2B? Is that the mean or median value of a strain? How many replicates are there for each mutant? Please state the information in the legends.

College of Science and Mathematics

Arkansas State University

Jonesboro, AR 72467

Tel: 870-972-3082

Email: rshields@astate.edu

November 8, 2023

Re: Response to Reviewer's Comments for Spectrum03149-23

Dear Dr. Jose Lemos,

We thank the reviewers for their comments on the manuscript and have edited the manuscript to address their concerns.

A point-by-point response follows:

Response to Reviewer #1:

Comment 1: Essential genes are genes that an organism requires to survive under specific conditions. In this study, the essential gene functions can be induced-terminated. The identified genes are growth-associated or growth-supporting but not essential since the genes are not completely knocked out, and the growth is not entirely forbidden. As the author also mentioned, the leaking function of CRISPRi may work to make the organism survive. Gene knock-out should be performed with several positive controls to confirm the results.

Response. Our reason for targeting essential/growth supporting genes with CRISPRi stems from prior research using Tn-seq. We realized that this was not introduced, and we have added a brief description starting at Line 78: "Our basis for targeting these genes with a CRISPRi system was conceived from prior transposon sequencing (Tn-seq) efforts exploring gene essentiality in *S. mutans* (17)." Note, that during our work with Tn-seq we validated >10 growth supporting genes using gene mutagenesis.

Comment 2: How can you evaluate the target-off effect of CRISPRi? Do the results are validated through gene expression or protein expression levels?

Response. We added text describing validation of the CRISPRi system that was conducted in Shields et al., 2020 (RNAseq, sgRNA design rules etc). We also added a sentence describing efforts to reduce off-target effects, and provided a reference which explores off-targets effects – so that readers are aware of this phenomenon. Text: "Validation of this CRISPRi system included determining repression with green fluorescent protein (GFP) readouts, measuring the effect of CRISPRi on the *S. mutans* transcriptome, and investigating sgRNA target-binding rules (16). When designing each sgRNA, care was taken to avoid potential off-target effects that are known to occur with CRISPR-based technologies (17)."

Comment 3: Line 39-40, one more reference could be added here (<https://www.ncbi.nlm.nih.gov/pmc/articles/PMC3216606/>).

Response. Added the *Streptococcus sanguinis* SK36 reference.

Comment 4: Line 84, the function of perR gene in Kajfasz's study should be described in the introduction.

Response. The following sentence was added after the first mention of the *perR* truncation: “PerR is a transcriptional regulator that responds to oxidative stress, and the mutation of *perR* in our lab-adapted strain of *S. mutans* likely confers an advantage in aerobic conditions.”

Comment 5: Line 106, the method to calculate the AUC_{sgRNA}/AUC_{control} values and how to get the p-values should be stated. Multiple testing corrections of the p-values should be performed.

Response. The current description of how we calculate the AUC at Line 112 is: “Plates were incubated in a microplate reader (Synergy 2) at 37 °C for 16 h, with OD₆₀₀ measurements recorded every 30 min. Afterwards, the area under the curve (AUC) was calculated for each well and compared to the average AUC of control strains.” We feel that this is an adequate description of how the AUC is calculated and it is self-explanatory that AUC_{sgRNA}/AUC_{control} is the AUC for an essential/growth-supporting sgRNA divided by the AUC for the control strains (these strains are described in the paragraph above (w/o sgRNA, sgRNA^{gtfB} and sgRNA^{scrB}). During our review of this we realized that we were not using a Student’s t-test (assuming equal variance) but instead a Welch’s t-test (assuming unequal variance), and have changed the text accordingly. We do not think that we need to have a P-value correction (e.g., Bonferroni) as it was always our intention to compare the control vs. an sgRNA targeting an essential or growth-supporting gene (we are not testing every possible comparison); setting a 5% false-positive rate seems fair. It is also important to stress, as with any screen, confirmation of phenotypes is required (for example as shown with Figure 4).

Comment 6: Line 110, please add an SI table to show the AUC_{sgRNA}/AUC_{control} values of the 141 strains.

Response. We added the raw data for BHI, CDM-Xylose, NaF and CPC microplate growth experiments (Table S1).

Comment 7: Line 133, are the comparisons shown in Fig 2A between w/o xylose and 0.5% (or 1%) xylose of the sgRNA^{gtfB} strain? What are the p-values of the comparison between w/o xylose and 0.5% (or 1%) xylose in the sgRNA^{scrB} and w/o sgRNA strains?

Response. Correct, we compared the sgRNA^{gtfB} strain w/o xylose and when the higher concentrations of xylose were added.

1% w/o sgRNA vs. 1% sgRNA^{scrB} P value = 0.814

0.5% w/o sgRNA vs. 0.5% sgRNA^{scrB} P value = 0.532

0% w/o sgRNA vs. 0% sgRNA^{scrB} P value = 0.928

Comment 8: Line 138, please explain the rationale for the threshold setting as 15%. An outlier analysis could be performed in the w/o xylose data to figure out a better threshold. The same comment is for the '0.5 increase' at line 142.

Response: We removed the sentence about planktonic vs. biofilm growth as it is an over interpretation of the data. In addition, we omitted words like ‘expected’ and ‘unexpectedly’.

Comment 9: What does each dot mean in Fig. 1 and 2B? Is that the mean or median value of a strain? How many replicates are there for each mutant? Please state the information in the legends.

Response: We added the following to both Fig. 1, Fig. 2B, and Fig. 3: “Each condition was repeated in triplicate and each dot represents the mean value of the triplicates.”

Response to Reviewer #2:

Comment 1: Line 65: there should be a comma after "expected"

Response. A comma was added to Line 65.

Comment 2: Figure S1: The black text is nearly impossible to read in some of the dark backgrounds. Please make the background color lighter or more transparent and/or change the text to white in these cases to improve readability.

Response: We have improved the readability of Figure S1.

Comment 3: Figure 1: I would suggest adding a header/title above each volcano plot so the reader can easily and quickly figure out what they are looking at. The size of the text relative to the figure should also be increased for readability.

Response: We have increased the size of the text and added a header to each figure.

Comment 4: Figure 2: Again, the size of the text is way too small relative to the figure to be readable...I printed out the document and I can't read it at all...I have to zoom way in on the computer. Also, the text in all figures is pixelated and low quality when you do zoom in.

Response: The text size was increased for both A and B. The resolution of all the figures should be improved; they are 600 DPI in .tif format.

Comment 5: Figure 2: The only comparisons that are statistically significant the two shown with ***?? None of the others are (w/o sgRNA w/o xylose vs 0.5 xylose, for example)?? Please clarify this and make sure it is clear to the reader which comparisons in the figure are and are not statistically significant.

Response: All other comparisons that we tested are not significant. For clarification, we have stated this in the Figure 2 legend: "Only statistically significant comparisons are shown; all other comparisons were found to be not significant". Requested comparisons with P values are shown below:

w/o sgRNA vs. 0.5% w/o sgRNA P value = 0.117

w/o sgRNA vs. 1% w/o sgRNA P value = 0.376

Comment 6: Lines 141-143: This point is a bit confusing. Please explain what you mean here a little better, this is with or without xylose and what do you mean "by expected"?

Response: We removed the words expected and unexpectedly from our description of these results. We also removed the sentence about planktonic vs. biofilm growth as it was a confusing analysis.

Comment 7: Figure S3: Again, please add titles to the graphs. In the legend, I believe the panel letter should come before the text describing the panel, not after (titles will help clear that up, in any case). The graph legends are also confusing...the concentrations are for the antimicrobial added? Or inducer? Please make all this more clear.

Response: Figure S3 has been updated (titles etc). The title for each begins with "Basal repression (w/o xylose)", which hopefully clarifies that xylose was not added to these experiments. We added the following sentence to clear up any confusion about the antibiotic concentration that was tested: "The concentrations of each antibiotic are shown in the figure legend."

Comment 8: Figure S4A: The order of concentrations in the legend makes no sense, please rearrange so they are in order of increasing. Also please add titles. Also, these are in mM, while the main text discusses things in terms of µg/ml, please make them all consistent. Also, if by

125 mM you mean 125 µg/ml, this does appear to inhibit growth, and therefore is not a sub-MIC concentration, as described in the main text?

Response: We apologize for the units error on Figure S4A; this has been changed from mM to µg/mL. We cannot see where we describe 125 µg/mL as inhibiting growth – we called it sub-MIC, which is a fair description.

Comment 9: Figure S4B: So the concentration used in the main experiment was 0.25 µg/ml, but that is not one of the growth curves shown? Since there did appear to be a minor lag of growth with 0.15625 and certainly a lag with 0.3125, please explain how you justify calling 0.25 a sub-MIC concentration.

Response: There were inconsistencies in the sub-MIC values between the Bioscreen C growth system (Figure S4) and the Synergy plate reader (CRISPRi screen data). We amended the concentration of CPC to 0.25 µg/mL because it was a more reliable sub-MIC during the microplate experiments – strains were notably more sensitive to CPC during the microplate experiments.

Comment 10: Line 186: Why not test with higher NaF concentration where you might expect to see more impact and determine which genes are affected/impacted?

Response: We altered the wording of the results to account for NaF concentration: This suggests that at a concentration of 125 µg/mL there are no essential genes in our library that are sensitive to NaF. Pyruvate kinase (Plate 1, *pykF*), enolase (Plate 1, *eno*) and components of the F-ATPase (Plate 1, *atpF*, *atpC*, *atpA*, *atpD*; Plate 3, *atpB*) are in our CRISPRi library. Although NaF has been reported to alter the activity of these enzymes, our findings suggest that basal repression of these genes does not impact NaF tolerance at 125 µg/mL. Testing with a higher concentration of NaF (~200 µg/mL) might increase the sensitivity of CRISPRi strains to NaF. It is also possible that repression of more than one essential gene at the same time (in a multiplex format) might cause more significant defects.

Comment 11: Figure 3: Again, please increase the size of the text relative to the figure. Is 1742 *fabK*? *fabK* is called out in the main text on line 195, but doesn't appear to be labeled as such in this graph.

Response: We have increased the size of the text and added a header to each figure. We also renamed 1742 to *fabK*.

Comment 12: Line 240: add reference for Smu Δcas9 Pxyl-dcas9 strain.

Response. Added reference for the strain.

Comment 13: Please speculate on why you do not see reduced growth with the basal repression. You spend Figure 1A showing that there isn't reduced growth without *xyl*, but then suggest transcription of many genes is likely repressed enough to increase sensitivity to antimicrobials. I think this is worth a sentence or two to reconcile.

Response: We added the following discussion (starting Line 175): Note, there is a contrast in that basal repression does not cause growth defects in rich media (Figure 1A) but addition of sub-MIC antibiotics does. In *E. coli* and *B. subtilis* it has been suggested that essential gene products are produced in sufficiently large quantities that despite basal repression, in rich media, cells are buffered against significant deleterious effects (10, 11). However, addition of antimicrobials that target specific essential genes/pathways causes this buffer to be disrupted, and the result is a loss of growth.

Best,
Dr. Robert C. Shields

Assistant Professor of Microbiology

Re: Spectrum03149-23R1 (Construction of an arrayed CRISPRi library as a resource for essential gene function studies in *Streptococcus mutans*)

Dear Dr. Robert C. Shields:

Your manuscript has been accepted, and I am forwarding it to the ASM production staff for publication. Your paper will first be checked to make sure all elements meet the technical requirements. ASM staff will contact you if anything needs to be revised before copyediting and production can begin. Otherwise, you will be notified when your proofs are ready to be viewed.

Sincerely,
Jose Lemos
Editor
Microbiology Spectrum